# A Comparison of the Treatment Effects of a Risperidone Solution, an Equal Ratio of DHA/ARA, and a Larger Ratio of Omega-6 PUFA Added to Omega-3 PUFA: An Open-Label Clinical Trial

**DOI:** 10.3390/cimb47030184

**Published:** 2025-03-12

**Authors:** Kunio Yui, George Imataka

**Affiliations:** 1Department of Pediatrics, Graduate School of Medicine, Chiba University, Chiba 260-8677, Japan; 2Department of Pediatrics, Dokkyo Medical University, Tochigi 321-0293, Japan; geo@dokkyomed.ac.jp

**Keywords:** autism spectrum disorder, risperidone solution, omega-3 PUFA, docosahexaenoic acid, omega-6 PUFA arachidonic acid, ceruloplasmin

## Abstract

We aimed to assess the efficacy, safety, and pharmacokinetics of an oral risperidone solution and two types of supplementations with PUFAs. We assigned 39 participants with mild ASD (mean age ± standard deviation = 14.6 ± 6.0 years) to three treatment groups (each n = 13): RIS-OS; equal doses of 240 mg of omega-3 PUFA docosahexaenoic acid and omega-6 PUFA arachidonic acid (1:1) (aravita); and omega-6 precursor linoleic acid (480 mg) and omega-3 precursor alpha-linolenic acid (120 mg) (4:1) (awake). The primary outcome was the Autism Diagnostic Interview—Revised score. The secondary outcomes were the Social Responsiveness Scale (SRS) and Aberrant Behavior Check scores. The results of the linear mixed-effects model revealed that the RIS-OS group exhibited significant improvement in the SRS subscale scores of social motivation at weeks 8, 12, and 16 compared with the aravita and awake groups, as well as in the SRS subscale score of social mannerisms at weeks 12 and 16 compared with the aravita group. Moreover, the RIS-OS group showed a trend towards significantly lower plasma ceruloplasmin (Cp) levels. Their plasma insulin-like growth factor (IGF) levels were significantly higher at week 8 than in the subsequent weeks. The high Cp and IGF levels may be attributed to reduced neuroinflammation. These findings demonstrate, firstly, that reduced inflammation through increased anti-inflammatory proteins such as Cp and IGF has clinical effects on the motivation–reward system and mannerisms in patients with ASD through the amelioration of dopamine D2, 5-HT2a, and 5-HT2b dysfunction.

## 1. Introduction

Autism spectrum disorder (ASD) is characterized by early-onset social communication deficits and repetitive sensory–motor behaviors, as well as a strong association with genetic and environmental factors [1]. Despite its increasing prevalence, there remain few established pharmaceutical strategies for effectively treating social deficits in ASD [2]. Accordingly, there is a need to establish effective evidence-based treatments [3].

A recent review showed that risperidone improved repetitive behaviors, inappropriate language, social withdrawal, and behavioral problems; however, it induced adverse events [4]. Another study reported that risperidone considerably improved maladaptive behavior and social skills, as indicated by a decrease in the Vineland Adaptive Behavior Scale and the parent-rated Aberrant Behavior Checklist (ABC) Irritability subscale scores [5]. Another review showed that risperidone significantly improved the ABC and Childhood Autism Rating Scale (CARS) scores [6], while a more recent one showed that it improved ABC lethargy and inadequate speech subscores, but significantly increased weight gain and waist circumference [7]. A multicenter, randomized, double-blind trial [8] found that risperidone significantly reduced ABC irritability subscores [8]. Risperidone treatment significantly improved core ASD symptoms, including social impairment and repetitive stereotyped behaviors; however, it induced weight gain and increased appetite, fatigue, drowsiness, dizziness, and drooling [8].

An extensive study on risperidone treatment for children and adolescents with ASD indicated that it significantly improved the restricted, repetitive, and stereotyped patterns of behavior, interests, and activities of children with ASD but did not significantly improve their deficits in social interaction and communication [8]. This may be attributed to the use of an insensitive rating scale in the social domain [9]. Therefore, it remains unclear whether risperidone mainly improves social interactions in patients with ASD.

The Food and Drug Administration has approved indications for oral risperidone, including tablets or oral solutions [10]. Compared with conventional tablets, an oral risperidone solution has significantly increased bioavailability, suggesting that an oral risperidone solution allows for good toleration and cooperation among patients with psychotic disorders, as well as doctor satisfaction [11]. Therefore, we developed an oral risperidone solution for improving the two core ASD symptoms.

Accordingly, we aimed to investigate whether treatment with an oral risperidone solution improves social communication deficits in patients with ASD. For this purpose, we used the parent version of the Social Responsiveness Scale (SRS), which has demonstrated adequate reliability and validity [12] in the assessment of the severity of ASD-related social impairment [13] or social communication deficits [14].

Polyunsaturated fatty acid (PUFA) administration has been shown to improve social and behavioral symptoms in patients with ASD through the role of omega−3 and omega−6 fatty acids in the dynamic regulation of synaptic transmission and the modulation of neurotransmitter release [15]. Docosahexaenoic acid (DHA) and arachidonic acid (ARA) are involved in numerous human physiological processes, including signaling pathways, gene expression, the structure and function of cellular membranes associated with G proteins, and gene expression [16]. DHA and ARA are deposited in the lipids of cell membranes that form gray matter, accounting for 25% of the total content of brain fatty acids. DHA and ARA are crucially involved in the protection against neurodegenerative pathologies [16]. Specifically, ARA is crucially involved in neurotransmitter release and neuroinflammation [17]. A previous double-blind, placebo-controlled, randomized trial indicated that large doses of ARA added to DHA improved social impairment in young individuals with ASD [18]. Taken together, treatment with DHA and ARA may crucially contribute towards improving the core social and behavioral symptoms of ASD.

There have been limited clinical trials comparing the therapeutic effects of an oral risperidone solution and large doses of ARA added to DHA on social impairment and behavioral symptoms in patients with ASD. This study aimed to compare the clinical effects of three treatments: an oral risperidone solution; equal doses of 240 mg of omega-3 DHA and omega-6 ARA (1:1); and omega-6 linoleic acid (480 mg) and omega-3 alpha-linolenic acid (120 mg) (4:1).

## 2. Materials and Methods

### 2.1. Participants

We included 39 individuals (30 male and 9 female; mean age ± SD = 14.6 ± 6.0 years) who were diagnosed with ASD according to the patient version of the Structured Clinical Interview for Diagnostic and Statistical Manual of Mental Disorder, Fifth Edition (DSM-V) [19], which was corroborated by a standardized semi-structured interview using the Autism Diagnostic Interview—Revised (ADI-R) [20]. Each diagnosis was established at screening based on an independent consensus between a psychiatrist (KY) and pediatrician (GI). This single-center, open-label, randomized controlled trial evaluated the efficacy of (1) equal doses of DHA and ARA (each 40 mg/capsule; aravita; n = 13), (2) relatively higher doses of omega-6 PUFA (linolenic acid 480 mg/capsule plus α-linolenic acid 120 mg/capsule; awake; n = 13), or (3) oral risperidone solution (RIS-OS) (n = 13; mean age ± SD = 15.5 ± 7.4 years). Participants were recruited from the Research Institute of Pervasive Developmental Disorders, Ashiya University (Ashiya, Japan) and the Child and Adolescent section of the Fujimoto Clinic (Kobe, Japan) between October 2009 and January 2012 through a local advertisement. Two physicians (a psychiatrist and pediatrician) were blinded to the group allocation. The research staff, who were responsible for enrolling the patients and their parents, conducted the study assessments, and thus were not blinded to the group allocation or study outcomes.

Physical examinations (sitting blood pressure, heart rate, weight, and height), motor function assessments (muscle tone, deep tendon reflex, and gait), and clinical laboratory tests (clinical chemistry, platelet activation, hematology, and urine toxicology) were conducted by a psychiatrist and pediatrician at screening and at the end of the trial.

The inclusion criteria were as follows: (1) absence of clinical or comorbid psychiatric disorders; (2) body weight ≥ 16 kg; (3) a verbal or performance intelligence quotient (IQ) > 70 at baseline based on the Wechsler Intelligence Scales for children and adolescents aged 6–16 years (Nader et al., 2015) [20] and that for adults [21], since high-functioning ASD is indicated by a total IQ score ≥ 70 [22]; and (4) a baseline score > 10 on the ABC subscale of social withdrawal [23], since the average score of this subscale in participants with neuropsychiatric diseases (e.g., deafness, epilepsy, cerebral palsy, and psychoses) aged 5–30 years was <10 [24].

We excluded participants who were expected to require hospitalization for behavioral symptoms; those with abnormal laboratory test findings; and those who had been treated with antidepressants, anxiolytic medication, or neuroleptics within the previous 5 months. Treatment of attention deficit/hyperactive disorder symptoms with stimulant medications was allowed during the study, as long as the patient’s dosage was stable for ≥3 months before the study and remained stable during the study. Additionally, we excluded participants who had participated in a program that taught appropriate social skills or had received parent-mediated communication-focused treatment within 3 months before the study.

The study protocol was approved by the ethics committee of the Fujimoto Clinic. Moreover, written informed consent was obtained from the participants, their parents, or both. The study was conducted in accordance with the principles of the Declaration of Helsinki. Clinical monitors examined the source documents to assess protocol adherence; additionally, all collected data were reviewed for protocol deviations.

### 2.2. Study Medication

In this open access clinical trial, RIS-OS 0.5–1.0 mg/mL was obtained from Ping An-Shionogi Co., Ltd. (Shanghai, China); aravita (DHA 240 ng and ARA 240 mg) was obtained from Suntory Wellness Ltd., Tokyo, Japan; and awake (linolenic acid 480 mg and α-linolenic acid 120 mg/capsule) was obtained from Sirius, Co., Ltd., Tokyo, Japan.

### 2.3. Study Design

This single-center, open-label, blinded-endpoint trial included three treatment groups comprising patients with moderate core behavioral and social symptoms of ASD; further, there was no placebo-treated control group. Patients, investigators, raters, and statisticians were not blinded to the treatment assignment, and all participants received the study intervention.

The outcome measures were the patient-/parent-reported effects of the three treatments on the ASD symptoms based on the ABC and SRS scores. Patients were monitored for adverse reactions during each session. To reduce bias, clinical effects were assessed after 1 week. Moreover, we assessed for a ≥25% reduction in the total ABC and SRS scores at each trial stage (4, 8, 12, and 16 weeks).

Since the included participants did not receive other medical drugs for ≥6 months before the study, there was no need to provide urine and blood samples for testing for drugs of abuse (including benzodiazepines, opiates, and cannabinoids). There were no cases of treatment discontinuation due to adverse events (tolerability) or for any other reason (acceptability).

After screening, 39 eligible participants were randomly allocated to the three aforementioned groups. Participants were not blinded, and nor was anyone else, at each research step (recruitment, intervention, and analysis).

Patients in the aravita group received six capsules daily, each containing 40 mg ARS and 40 mg DHA, yielding a total daily dose of 240 mg each of ARA and DHA. The awake group received two capsules daily, each containing omega-6 linolenic acid 480 mg and omega-3 α-linoleic acid 120 mg, yielding a total daily dose of 960 mg and 240 mg for linoleic acid and α-linoleic acid, respectively. Patients in the RIS-OS group received 0.5–1.0 mg of RIS-OS, with participants aged ≥13 years receiving a dose of 1.0 mg (Table 1).

### 2.4. Outcomes

The primary outcome measure was the ADI-R score, which assesses parent-reported autistic behaviors and social communication patterns [25]. Secondary outcomes included the SRS and ABC scores. Outcomes were assessed at baseline and at subsequent 4-week intervals (4, 8, 12, and 16 weeks). The SRS is a quantitative scale used to assess the presence and extent of autistic social impairment and social communication; moreover, it is correlated with ADI-R scores. SRS scores reflect numerous components of social interactions (visual stimuli, reading emotional cues, and auditory information) related to the learning of social cues and integration of appropriate social responses [26].

The ABC was primarily intended for treatment evaluation in psychopharmacological and behavioral intervention trials [27] on children and adolescents with high-functioning ASD. It comprises 58 items, each scored on a 4-point scale. The items fall into five subscales: (1) irritability, (2) social withdrawal, (3) stereotype, (4) hyperactivity, and (5) inappropriate speech [28]. Both the SRS and ABC subscales were completed by the parent.

### 2.5. Safety Assessment

Safety monitoring was conducted throughout the study. Physical examinations and laboratory tests (e.g., clinical chemistry and hematology) were performed at baseline and the final visits. Adverse events were assessed using the Common Terminology Criteria for Adverse Events (CTCAE) Version 5.0 [29], which focused on the known side effects of ARA and RIS-OS using open-ended questions. The side effects included nausea, vomiting, stomach aches, and diarrhea. The relationship between each adverse event and study intervention was assessed at the time at which the event was reported. No adverse events of any grade were reported in any of the three treatment groups.

Assays of plasma levels of PUFAs, superoxide dismutase, ceruloplasmin, transferrin, and IGF were performed.

### 2.6. Blood Sampling Procedures

Blood samples obtained under fasting conditions were used to assess the beneficial effects of EPA and DHA supplementation [30]. Whole blood samples were collected after a 3 h fasting interval. Regarding the fasting interval, the extent of return to the fasting state at 2 h after a glucose challenge has been negatively associated with the risk of incident diabetes mellitus in normoglycemic individuals. Therefore, we considered a 3 h fasting period after breakfast to be appropriate for blood sampling. Blood samples were collected into EDTA tubes by venipuncture and immediately placed on ice. Plasma was obtained by centrifugation for 25 min at 3000× *g* at room temperature (24 °C). To reduce the effects of circadian variations, blood sampling was performed between 1100 h and 1230 h in a quiet laboratory room. The blood samples were frozen at −80 °C until subsequent analysis for plasma PUFAs, ceruloplasmin (Cp), transferrin (Tf), superoxide dismutases (SOD), and insulin-like growth factor (IGF) levels by specialists at SRL, Inc. (Tokyo, Japan).

#### 2.6.1. Plasma Levels of PUFAs

We analyzed the fatty acid composition of the total phospholipid fraction in the plasma as previously described (Hamazaki et al., 2006). Briefly, total lipids were extracted from the plasma as described by Bligh and Dyer (Blight et al., 1959). After transmethylation with HCL-methanol, the PUFA composition was analyzed through gas chromatography (GC2010, Shimazu Co., Tokyo, Japan). The sensitivity of our measurement method for plasma DHA and ARA levels was 0.2 μg/mL. The intra- and inter-assay coefficients of ARA were 110.14 μg/mL and 100.63 μg/mL, respectively, while those of DHA were 73.87 μg/mL and 68.07 μg/mL, respectively. The plasma level of each PUFA was expressed as the weight% of total fatty acids (Table 2).

#### 2.6.2. Antioxidant Proteins

Increased oxidative stress may contribute to the pathogenesis and clinical symptoms of ASD [31].

#### 2.6.3. Ceruloplasmin

CP is a potent antioxidant protein [32] that is crucially involved in neuropathological conditions by stimulating pro-inflammatory and neurotoxic molecules in microglia [33]; additionally, it is closely related to neuroinflammation [34]. ASD is associated with high plasma copper and Cp levels [35].

Plasma Cp levels were measured using a Bering BN II nephelometer (Siemens Healthcare Diagnostics, Kabushiki Kaisha, Minato Ward, Tokyo, Japan), which has a sensitivity of 3.0 mg/dL. The intra- and inter-assay coefficients were 10.2 mg/dL and 10.1 mg/dL, respectively.

#### 2.6.4. Transferrin

Tf is an antioxidant protein [36]. Further, Tf is an iron transport protein in the redox-safe state [37]. Plasma TF levels were assessed using a standard turbidimetric assay with an automated biochemical analyzer (JCA-BM8000 series, JEOL Ltd., Tokyo, Japan). The minimum detectable concentration was 21.0 mg/dL. The intra- and inter-assay coefficients were 108.1 mg/dL and 107.4 mg/dL, respectively.

#### 2.6.5. Superoxide Dismutase

SOD is a group of metalloenzymes that provide an important antioxidant defense mechanism that is required to preserve body reactive oxygen species (ROS) levels. SOD has therapeutic potential against various diseases related to deficient ROS levels [38].

Plasma SOD levels were estimated based on the decreasing rate of nitrite produced by hydroxylamine and superoxide anions according to the nitrite method using VERSA Max (Milecular Devoce Ltd., Tokyo, Japan). Hydroxylamine or its O-sulfonic acid, xanthine oxidase, hypoxanthine, EDTA, and the plasma sample were incubated with or without KCN at pH 8.2, 37 °C, for 30 min. A diazo dye-forming reagent was added; moreover, absorption was measured at 550 nm. Plasma SOD levels were expressed as U/mL. The assay sensitivity was 0.3 U/mL. The intra- and inter-assay coefficients were 2.11 U/mL and 2.10 U/mL, respectively.

#### 2.6.6. Plasma IGF Levels

Plasma IGF-1 levels were measured using an immunoradiometric assay with commercially available kits (TFB, INC. Tokyo, Japan) [39].

### 2.7. Statistical Analyses

The sample size was calculated using G*Power 3.1.2 with consideration of an effect size of 0.70, statistical power of 80%, and statistical significance of a two-tailed *p* value of 0.05. The sample size was estimated using GPower 3.1.2 with a power of 80%, an alpha of 0.05, and an expected effect size of 0.7 based on previous studies on ASD treatments.

Statistical analyses were performed using a linear mixed-effects model (LMM) [40], which is a common and powerful tool for analyses of data with complex correlated structures and multiple sources of variation. An individual parameter in the LMM provides a random effect. A fixed effect provides an average effect of a specific experimental factor. This method explicitly incorporates both fixed and random effects, with random effects accounting for unwanted sources of variation in the respective measurements. LMM fitting has demonstrated utility and flexibility in the identification of metabolites that are significantly affected by important factors [41].

Given the non-normal data distribution, between-group comparisons at baseline and 16 weeks were performed using the Mann–Whitney U test. Arson’s chi-square test is widely used to test the goodness of fit between categorical data and a given discrete distribution function [42]. Statistical analyses were performed using SPSS version 26.

## 3. Results

### 3.1. Patient Characteristics

Table 1 summarizes the patient characteristics. There were no among-group differences in baseline characteristics. The included participants met the DSM-V criteria for autistic disorder, including delays in language as used in communication or symbolic play (n = 7); qualitative impairments in social interaction and communication (n = 28); and restricted, repetitive, and stereotyped patterns of behavioral interest or activities (n = 21). As shown in Table 1, all participants met the three ADI-R three (domain A: Reciprocal Social Interest; domain B: Abnormalities in Communication; and domain C: Restricted, Repetitive, Stereotyped Pattern of Behaviors). The average scores for domain A, B, and C were 19.38 ± 6.27, 10.5 ± 3.72, and 7.87 ± 5.0, respectively, with these scores indicating mild-to-moderate ASD [43].

There were no significant among-group differences in age (U = 20.0; *p* = 0.95) or mean baseline scores of the SRS and ABC subscales (all *p* values > 0.05) (Table 1). The total SRS and ABC scores at baseline in the aravita group were 88.15 ± 30.88 and 49.53 ± 29.14, respectively; those in the awake group were 118.8 ± 24.8 and 67.8 ± 28.7, respectively; and those in the RS-OS group were 96.8 ± 14.8 and 50.9 ± 30.31, respectively. Total SRS scores of ≤59, 60–65, 66–75, and ≥76 indicate normal, mild, moderate, and severe social impairment, respectively [44]. Accordingly, the total SRS and ABC scores at baseline in the RIS-OS and awake groups were lower and slightly higher, respectively, than those reported for ASD (Table 1); however, the total SRS scores in both groups indicated impaired social reciprocity. Therefore, all participants were diagnosed with mild core ASD symptoms.

### 3.2. Efficacy Results

There was no difference in the treatment effect according to age group. The LMM results revealed significantly improved SRS subscale scores for motivation at 4, 8, 12, and 16 weeks compared with the aravita and awake groups. Additionally, the awake group showed significantly improved SRS subscale scores for motivation compared with the aravita group at week 16 (Table 2 and Table 3).

Pearson’s chi-square test (χ^2^ test) revealed a significantly higher number of patients with a 25% improvement in the SRS total scores in the RIS-Os group than in the aravirta group. However, there was no significant between-group differences in the number of patients with a 50% improvement in the SRS and ABC total scores.

### 3.3. Plasma Antioxidant Protein Levels

The RIS-OS group showed a trend towards significantly lower plasma Cp levels than the awake group at 8 weeks (*p* = 0.058) and 16 weeks (*p* = 0.086). Moreover, plasma Cp levels were significantly lower in the risperidone group than in the aravita group. Therefore, our findings indicate that decreased plasma Cp levels may contribute to improving the SRS subscale scores for mannerisms and motivation (Table 4).

Additionally, plasma IGF levels were significantly higher in the RIS-OS and awake groups than in the aravita group (Table 5).

### 3.4. Adverse Events

No adverse events were reported in the three groups. Safety endpoints included conditioning regimen-related safety, short- and long-term safety of the three treatments, and monitoring of adverse events and laboratory values.

## 4. Discussion

Our findings indicated that the RIS-OS group showed significantly improved SRS subscale scores for cognition at week 8 and for motivation at weeks 4, 8, and 12 compared with the aravita and awake groups (Table 2). Moreover, there was a significantly higher number of patients with a 25% improvement in the total SRS scores in the RIS-Os group than in the aravirta group.

The social motivation hypothesis proposes that individuals with ASD have deficits in the processing of social reward systems [45]. The social motivation theory indicates an association between deficits in the reward system and the processing of critical social stimuli [46]. Several clinical studies have indicated an association between motivation and social reward anticipation [47]. Children with ASD may be less active in the processing of rewards [48], and patients with ASD have impaired social reward processing [49]. ASD is a social motivational disorder characterized by less attention being paid to the social environment and less pleasure being experienced through social rewards [50].

The mechanisms underlying impaired reward responsiveness in ASD may involve dysfunction of the dopaminergic–oxytocinergic ‘wanting’ circuitry, including the ventral striatum, amygdala, and ventromedial prefrontal cortex [51]. The anterior cingulate and orbital cortices, as well as the ventral striatum, process different aspects of reward evaluation [52]. Furthermore, dopamine and opioids in the medial preoptic nucleus and ventral tegmental area play distinct roles in motivation and reward [53], while the subcortical reward and motivation systems are related to limbic cortical region processes [54]. The social motivation hypothesis proposes that patients with ASD who have deficits in the social reward system engage less in social interactions [55]. Alterations in the dopaminergic reward system contribute to motivation problems in ASD [56]. Our findings indicated that RIS-OS significantly improved motivation. Reward system deficits and social motivation are associated with the processing of critical social stimuli [47].

Dopamine-mediated brain activity is intimately linked to reward-driven cerebral responses; moreover, the D2 antagonist risperidone influences dopamine-mediated reward function via potential vascular confounders of the reward-related blood oxygen level-dependent signal, which may be advantageous when investigating drug action in the central nervous system [57]. RIS has demonstrated 5-HT_1A_ and 5-HT_2A_ antagonism, as well as D_2_R antagonism [58]. Moreover, 5-HT_1A_ and 5-HT_2A_ may be encoders of rewards and be involved in the acquisition and extinction of reward encoding [59]. Therefore, RIS-related 5-HT_1A_ and 5-HT_2A_ receptors may contribute to motivation via the reward system. Accordingly, treatment with the D2 antagonist, RIS-OS, may significantly improve the processing of rewards and motivation. Therefore, 5-HT_1A_ and 5-HT_2A_ receptors may be related to the reward system.

To our knowledge, this is the first study to propose that the association between social motivation and the reward system contributes to the pathophysiology in young individuals with ASD through dysfunction of dopamine D2, 5-HT_1A_, and 5-HT_2A_ receptors.

In our study, the RIS-OS group showed significantly improved scores on the SRS subscale of mannerism at weeks 12 and 16 compared with the aravita group. Mannerisms are maladaptive behaviors related to withdrawal or aggressive behavior due to a disturbed perception of social signals [60]; further, they are related to the D2/D3 receptor [61]. Therefore, RIS-OS may improve these mechanisms via dopamine D2 dysfunction, as well as 5-HT_1A_ and 5-HT_2A_ receptors. In addition, the awake group exhibited significantly improved SRS subscale scores for motivation at week 16. This could be attributed to the effect of endocannabinoids, which are oxidation-independent ARA derivatives that are important for brain reward signaling and motivational processes [62]. Taken together, our findings suggest that RIS-OS improves the impaired reward system, resulting in improvements in the SRS subscales of motivation and mannerisms. However, there was no association between the ABC subscale scores and the treatment effects on the motivation–reward systems.

Our findings suggest that plasma Cp levels contribute to ASD symptoms. Decreased serum Cp levels have been positively correlated with a decrease in dopamine transporter density in patients with Parkinson [63]. Cp acts as a defense mechanism against inflammation [64]. Neurodevelopmental disorders such as ASD are closely related with neuroinflammation [65]. Moreover, high Cp levels in the cerebrospinal fluid are associated with accelerated cognitive decline [66], and ASD is associated with high plasma levels of copper and Cp [36].

In our study, plasma IGF levels at week 8 were significantly higher in the RIS-OS and awake groups than in the aravita group. RIS has been used as risperidone or PUFA supplementation for ASD treatment, leading to an increase in plasma IGF levels [67]. IGF has anti-inflammatory effects on astrocytes and microglia due to the inhibition of blood–brain barrier permeability [67]; further, IGF levels are positively correlated with the ARA precursor linoleic acid [68]. Therefore, the presence of high linoleic acid levels may increase plasma IGF levels [69]. In addition, since RIS exhibits anti-inflammatory activity [70], it promotes the inflammatory effects of plasma IGF levels. This is the first study to reveal that plasma IGF levels are associated with RIS-OS and are positively associated with the plasma levels of linoleic acid.

Plasma Cp levels may be associated with ASD symptoms. Plasma Cp levels were low in the RIS-OS group at week 8, which may reflect improved neuroinflammation. Serum Cp levels have been shown to significantly decrease with the improvement of microinflammation during therapeutic albumin infusion [71] due to the inhibition of tetrethiomolybdate [72].

A study using the SRS subscale scores reported an association between motivation and mannerisms, mediated by blood lead concentrations in children with ASD aged 7–8 years [73]. Adolescents who were assigned male at birth had significantly higher scores on SRS in the Total Scale and the Social Motivation and Autistic Mannerisms subscales compared to the female control group [74]. Therefore, the SRS subscale scores for motivation and mannerisms may be related to similar brain functions.

In our study, the awake group showed significantly higher IGF levels at week 8 than the aravita and RIS-OS groups. Linoleic acid is an ARA metabolite, which induces the expression of the antioxidant manganese SOD gene via ROS, which are derived from ARA metabolism [75,76]. Therefore, the high levels of the ARA precursor linoleic acid in the treatment administered to the awake group induced significantly higher serum SOD levels.

This study included relative small samples and lacked a placebo control group. There is therefore a need for further research.

## 5. Conclusions

Our findings indicated that social motivation and mannerisms were significantly improved in the RIS-OS group compared with the aravita and awake groups. Further, the RIS-OS group showed a trend towards significantly lower plasma Cp levels compared with the awake and aravita groups at 8 and 16 weeks. Accordingly, our findings indicate that lowered plasma Cp levels may contribute to improving the SRS subscale scores for mannerisms and motivation.

## Figures and Tables

**Table 1 cimb-47-00184-t001:** Baseline patient characteristics of three groups.

Group	Aravita (n = 13)	Awake (n = 13)	RIS-OS (n = 13)
Age (Years)	14.6 ± 6.2	9.8 ± 2.8	13.4 ± 7.2
Sex (Male/Female)	10: 3	9: 4	11: 2
ADI-R			
A: Social Interaction	25.9 ± 4.0	11.6 ± 2.5	10.5 ± 3.6
B: Abnormal Communication	18.4 ± 3.6	11.1 ± 2.8	9.3 ± 4.7
C: Repetitive Stereotyped Behavior	13.9 ±3.9	8.8 ± 4.9	3.9 ± 4.6
Social Responsiveness Scale (SRS)			
Awareness	15.9 ± 4.2	12.1 ± 3.3	9.9 ± 4.0
Cognition	25.0 ± 5.7	24.3 ± 4.7	19.4 ± 5.3
Communication	46.8 ± 1.3	42.7 ± 10.6	32.4 ±11.3
Motivation	24.4 ± 7.0	21.4 ± 4.2	16.4 ± 4.8
Mannerisms	23.2 ± 6.5	18.5 ± 5.4	17.0 ± 8.4
Total Score	137.4 ± 28.9	116.8 ± 24.7	95.1 ± 28.5
Aberrant Behavior Checklist (ABC)			
Irritability	13.9 ± 9,4	14.3 ± 7.3	13.2 ± 9.0
Social Withdrawal	29.5 ± 7.3	22.2 ± 8.7	13.4 ±9.5
Stereotyped Behavior	7.2 ± 6.2	6.3 ± 5.2	4.0 ± 4.1
Hyperactivity	20.5 ± 12.2	23.2 ± 10.8	15.2 ±10.2
Inappropriated Speech	5.1 ± 4.0	4.1 ± 2.8	4.1 ± 2.8
Total Scores	76.3 ± 33.2	67.9 ± 28.6	50.2 ± 30.1

**Table 2 cimb-47-00184-t002:** The SRS subscale scores of motivations.

Time	Arm1	Arm2	Estimate	Probt	Probability	
					Lower	Upper
4 weeks	Awake	Aravita	−6.09	0.0004 *	−9.386	−2.774
4 weeks	Awake	Ris-OS	1.67	0.334	−1.724	5.067
4 weeks	Aravita	Ris-OS	7.75	<0.001 *	4.168	11.334
8 weeks	Awake	Aravita	−4.92	0.0004 *	−8.242	−1.801
8 weeks	Awake	Ris-Os	3.97	0.0022 *	0.583	7.375
8 weeks	Aravita	Ris-Os	8.90	<0.001 *	5.322	12.488
12 weeks	Awake	Aravita	−6.00	0.0005 *	−9.318	−2.687
12 weeks	Awake	Ris-Os	3.38	0.052	−0.032	6.759
12 weeks	Aravita	Ris-Os	9.36	<0.001 *	5.784	12.949
16 weeks	Awake	Aravita	−3.93	0.021	−7.242	−0.619
16 weeks	Awake	Ris-Os	2.98	0.08	−0.417	6.375
16 weeks	Aravita	Ris-Os	6.95	0.0002 *	3.322	10.487

* Statistically significant.

**Table 3 cimb-47-00184-t003:** The SRS subscale scores of mannerisms.

Time	Arm1	Arm2	Estimate	Probt	Probability	
					Lower	Upper
4 weeks	Awake	Aravita	−5.110	0.027 *	−9.614	0.606
4 weeks	Awake	Ris-OS	−0.289	0.89	−4.673	4.095
4 weeks	Aravita	Ris-OS	4.820	0.04 *	0.217	9.425
8 weeks	Awake	Aravita	−2.110	0.34	−6.614	2.393
8 weeks	Awake	Ris-Os	−0.0058	0.98	−4.442	4.325
8 weeks	Aravita	Ris-Os	2.051	0.37	−2.552	6.655
12 weeks	Awake	Aravita	−3.264	0.25	−7.767	1.240
12 weeks	Awake	Ris-Os	1.480	0.50	−2.904	5.863
12 weeks	Aravita	Ris-Os	4.744	0.046	0.139	9.348
16 weeks	Awake	Aravita	−1.649	0.47	−6.615	2.855
16 weeks	Awake	Ris-Os	3.018	0.17	−1.366	7.402
16 weeks	Aravita	Ris-Os	4.669	0.047	0.063	9.271

* Statistically significant.

**Table 4 cimb-47-00184-t004:** Plasma Cp levels.

Time	Arm1	Arm2	Estimate	t-Value	Probt	Probability	
						Lower	Upper
8 weeks	Aravita	Awake	1.915	0.981	0.334	−2.054	5.895
8 weeks	Aravita	Ris-Os	3.776	1.983	0.058	−0.136	7.669
8 weeks	Awake	Ris-Os	1.850	0.942	0.352	−2.131	5.832
16 weeks	Aravira	Awake	3.545	1.762	0.086	−0.615	7.424
16 weeks	Aravita	Ris-Os	0.920	0.480	0.635	−2.983	4.824
16 weeks	Awake	Ris-Os	−2.534	−1.294	0.205	−6.516	1.148

**Table 5 cimb-47-00184-t005:** Plasma IGF levels.

Time	Arm1	Arm2	Estimate	t-Value	Probt	Probability	
						Lower	Upper
8 weeks	Aravita	Awake	−74.811	−2.482	0.023 *	−137.990	−11.630
8 weeks	Aravita	Ris-Os	−14.401	−2.683	0.015 *	−158.850	34.565
8 weeks	Awake	Ris-Os	−89.212	−0.622	0.545	−63.367	34.564
16 weeks	Aravira	Awake	32.061	1.033	0.316	−32.120	94.242
16 weeks	Aravita	Ris-Os	−8.106	−0.352	0.499	46.682	92.591
16 weeks	Awake	Ris-Os	22.954	0.692	0.732	−57.072	40.859

* Statistically significant.

## Data Availability

The participants in this study did not give written consent for their data to be shared publicly, so due to the sensitive nature of the research, data are not available.

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
