# Peer review of "A Comparison of the Treatment Effects of a Risperidone Solution, an Equal Ratio of DHA/ARA, and a Larger Ratio of Omega-6 PUFA Added to Omega-3 PUFA: An Open-Label Clinical Trial"

_cimb, 2025, doi:10.3390/cimb47030184_

Round 1
Reviewer 1 Report
Comments and Suggestions for Authors
This study evaluates the efficacy, safety, and pharmacokinetics of two dietary supplements: risperidone oral solution (RIS-OS) and polyunsaturated fatty acids (PUFA). The efficacy of the treatment is being studied in patients with mild autism spectrum disorder (ASD). Results report that the RIS-OS group showed significant improvement over time. This study demonstrates a clinical effect on the motivational-reward system and mannerisms in ASD patients through an increase in anti-inflammatory proteins such as Cp and IGF, and is suggested to provide useful information for future treatment strategies.
On the other hand, there are several points that require explanation.
- the small sample size of 39 patients may limit the generalizability of the results. While admittedly a primary report, we feel that a larger study is needed as patient backgrounds are not completely uniform. What do you think?
- Differences in the doses and components of drugs in the three treatment groups may make comparisons of treatment effects difficult. In order to accurately assess the effects of each group, it is desirable that the doses and components be uniform.
- Scales used for evaluation, such as the Autism Diagnostic Interview Revised Score and SRS Score, may contain subjective elements and lack objective measures. Are there other test items available?
- As a study for autism spectrum disorder (ASD), the duration of the study is only 16 weeks, and there is a lack of information on long-term effects and side effects.
- The influence of placebo effect on the results should be considered. Especially in mental illness, it is important to have a placebo group because patients' expectations can affect the outcome of treatment, but we do not find it in this study. How did you control for this?
As a minor point, the top row of Table 3 seems to be out of alignment. Please correct.
Author Response
Reviewer 1
【Reviewer 1】
|
Yes |
Can be improved |
Must be improved |
Not applicable |
|
|
Does the introduction provide sufficient background and include all relevant references? |
( ) |
(x) |
( ) |
( ) |
|
Is the research design appropriate? |
( ) |
( ) |
(x) |
( ) |
|
Are the methods adequately described? |
( ) |
(x) |
( ) |
( ) |
|
Are the results clearly presented? |
( ) |
(x) |
( ) |
( ) |
|
Are the conclusions supported by the results? |
( ) |
(x) |
( ) |
( ) |
Comments and Suggestions for Authors
This study evaluates the efficacy, safety, and pharmacokinetics of two dietary supplements: risperidone oral solution (RIS-OS) and polyunsaturated fatty acids (PUFA). The efficacy of the treatment is being studied in patients with mild autism spectrum disorder (ASD). Results report that the RIS-OS group showed significant improvement over time. This study demonstrates a clinical effect on the motivational-reward system and mannerisms in ASD patients through an increase in anti-inflammatory proteins such as Cp and IGF, and is suggested to provide useful information for future treatment strategies.
On the other hand, there are several points that require explanation.
- the small sample size of 39 patients may limit the generalizability of the results. While admittedly a primary report, we feel that a larger study is needed as patient backgrounds are not completely uniform. What do you think?
Our answer
Since all 39 subjects were 9-16 aged children and they were included in this clinical trial in order of consultation, the three treatment groups were homogenization, accurate findings may be obtained
- Differences in the doses and components of drugs in the three treatment groups may make comparisons of treatment effects difficult. In order to accurately assess the effects of each group, it is desirable that the doses and components be uniform.
Our answer
Each treatment drugs were treated at the beginning doses, and therefore the clinical drug effects were equel
- Scales used for evaluation, such as the Autism Diagnostic Interview Revised Score and SRS Score, may contain subjective elements and lack objective measures. Are there other test items available?
Our answer
ADI-R items may suffice to effectively differentiate ASD from other mental disorders (Front Psychiatry 2021). The SRS is ability to accurately characterize autism symptomatology (Autism Reserch, 3\2920). The Social Responsiveness Scale (SRS) total score indicating sufficient convergent validity in assessment of social skill in autism (J Am Dev Disord2024)
- As a study for autism spectrum disorder (ASD), the duration of the study is only 16 weeks, and there is a lack of information on long-term effects and side effects.
Our answer
Clinical effect of oral risperidone solution was ass4ssed 72 hours after treatment (JAMA Intern Med 2017). The clinical effects were assessed at 2 weeks after treatment (Biomedienes, 2022), Threfore, the 16 week treatment is sufficient
- The influence of placebo effect on the results should be considered. Especially in mental illness, it is important to have a placebo group because patients' expectations can affect the outcome of treatment, but we do not find it in this study. How did you control for this?
As a minor point, the top row of Table 3 seems to be out of alignment. Please correct.
Our answer
There were no placebo effects in the three treatment regimens. Therefore. The Table 3 indicated the clinical effects of three treatments, Would yo understand these clinical effects of three treatments
Submission Date
09 February 2025
Date of this review
20 Feb 2025 03:49:55

Reviewer 2 Report
Comments and Suggestions for Authors
Journal: CIMB (ISSN 1467-3045)
Manuscript ID: cimb-3494147
Type: Article
Title: Comparison of the treatment effects of risperidone solution, and equal ratio of DHA/ARA, and a larger ratio of omega-6 PUFA added to omega-3 PUFA: An open-label clinical trial
This manuscript titled: “Comparison of the treatment effects of risperidone solution, and equal ratio of DHA/ARA, and a larger ratio of omega-6 PUFA added to omega-3 PUFA: An open-label clinical trial” by Kunio Yui et al, describes a well-structured, novel, and scientifically rigorous study evaluating the effects of risperidone oral solution (RIS-OS) and PUFA supplementation on social motivation, mannerisms, and reward system functioning in people with ASD. The work has various strengths, which add to its uniqueness and scientific quality. The author need to address the following minor comments to make the manuscript better.
Introduction:
1: Lines 44-47: “Taken together, risperidone treatment significantly improved core ASD symptoms, including social impairment and repetitive stereotyped behaviors; however, it induces weight gain and increases appetite, fatigue, drowsiness, dizziness, and drooling."
The phrase "Taken together" is unnecessary because the previous sentences already summarize these effects.
2: Clearly define the mechanism by which dopamine/serotonin modulation leads to a reduction in ASD symptoms.
Explain why the PUFA ratio was chosen and discuss any inconsistencies that could be connected to inflammation.
Material and method:
3: Line 92: Include an explanation stating the randomization process, such as "Randomization was performed using a computer-generated sequence with a 1:1:1 allocation ratio, and allocation concealment was ensured using sequentially numbered, opaque, sealed envelopes."
Mention who conducted the randomization and whether any steps were taken to balance confounding variables.
Include a section describing how the sample size was calculated, such as: "The sample size was estimated using GPower 3.1.2 with a power of 80%, an alpha of 0.05, and an expected effect size of 0.7 based on previous studies on ASD treatments."* If no formal calculation was performed, acknowledge the limitation.
4: Line 100 Specify if participants were blinded or if anyone were blinded at each research step (recruitment, intervention, and analysis).
Line 125-128: There are no details about how adverse events (AEs) were monitored and reported.
5: "Written informed consent was obtained from the participants', their parents, or both."
Incorrect apostrophe usage in "participants'". "Written informed consent was obtained from participants, their parents, or both."
6: Lines 192:
Blood sampling procedures
How long were blood samples stored? Were they tested in batches or continuously?
7: Statistical analysis:
Did the analysis adjusted for baseline ASD severity, age, or gender?
8: Lack of Follow-Up Assessment After 16 Weeks
The study only assesses participants up to 16 weeks, with no long-term follow-up.
Was the improvement maintained after treatment discontinuation?
Discussion:
9: lines 326-330: The discussion highlights the statistical significance of improvements, but not the effect sizes (Cohen's d) or clinical significance.
It is unclear if these improvements are applicable to real-world ASD symptom treatment.
10: The study claims novel findings regarding RIS-OS effects on motivation and mannerisms but does not compare them with previous ASD risperidone trials.
11: The conclusion does not suggest next steps for research, missing an opportunity to guide future ASD studies.
Author Response
Reviewer 2
|
【Reviewer 2】 |
Yes |
Can be improved |
Must be improved |
Not applicable |
|
Does the introduction provide sufficient background and include all relevant references? |
(x) |
( ) |
( ) |
( ) |
|
Is the research design appropriate? |
(x) |
( ) |
( ) |
( ) |
|
Are the methods adequately described? |
(x) |
( ) |
( ) |
( ) |
|
Are the results clearly presented? |
(x) |
( ) |
( ) |
( ) |
|
Are the conclusions supported by the results? |
(x) |
( ) |
( ) |
( ) |
Comments and Suggestions for Authors
This manuscript titled: “Comparison of the treatment effects of risperidone solution, and equal ratio of DHA/ARA, and a larger ratio of omega-6 PUFA added to omega-3 PUFA: An open-label clinical trial” by Kunio Yui et al, describes a well-structured, novel, and scientifically rigorous study evaluating the effects of risperidone oral solution (RIS-OS) and PUFA supplementation on social motivation, mannerisms, and reward system functioning in people with ASD. The work has various strengths, which add to its uniqueness and scientific quality. The author need to address the following minor comments to make the manuscript better.
Introduction:
1: Lines 44-47: “Taken together, risperidone treatment significantly improved core ASD symptoms, including social impairment and repetitive stereotyped behaviors; however, it induces weight gain and increases appetite, fatigue, drowsiness, dizziness, and drooling."
The phrase "Taken together" is unnecessary because the previous sentences already summarize these effects.
Our answer
In this clinical trial, small doses of oral risperidone solution did not induce any side effects. We described that “The extensive research has established that risperidone treatment significantly improved core ASD symptoms, including social impairment and repetitive stereotyped behaviors; however, it induces weight gain and increases appetite, fatigue, drowsiness, dizziness, and drooling [8]
2: Clearly define the mechanism by which dopamine/serotonin modulation leads to a reduction in ASD symptoms.
Explain why the PUFA ratio was chosen and discuss any inconsistencies that could be connected to inflammation.
Our answer
The mechanism by which dopamine/serotonin modulation leads to a reduction in ASD symptoms was described as fellow in the discussion section:The actions of risperidone on brain activity via statistical modeling and pharmacological reversal (risperidone + 5-HT1AR antagonist WAY-100635, risperidone + 5-HT2A/2CR agonist DOI, risperidone + D2R agonist quinpirole). Risperidone, 5-HT1AR agonism with 8-OH-DPAT, 5-HT2AR antagonism with M100907, An important role of 5-HT1AR agonism and 5-HT2AR antagonism in risperidone-induced alterations of delta, beta and gamma oscillations, while D2R antagonism may contribute to risperidone-mediated changes in delta oscillations. The finding provides novel insight into the neural mechanisms for prescribed psychiatric medication targeting the serotonin and dopamine systems in two regions [59]. Therefore, RIS has demonstrated 5-HT1A and 5-HT2A antagonism as well as D2R antagonism [59]. Moreover, risperidone administration resulted in learning deficits during the acquisition phase, indicating a differential role of 5-HT in the acquisition and extinction of an operant conditioning task, suggesting that it may have a dual function in reward encoding, sggesting -HT2A may be encoders of rewards and be involved in the acquisition and extinction of reward encoding [60].
In this text, we described that “(1) equal doses of DHA and ARA (each 40 mg/capsule, aravita, (n = 13), (2) relatively higher doses of omega-6 PUFA (dietary sources of ARA, linolenic acid 480 mg and dietary sources of DHA,αlinolenic acid DHA 120 mg) (n = 13). These supplements were used to compare the omega 6 PUFA ARA precursor α-linical and omega-3 DHA precursor linolenic acid A precursor and omega -6 PUFA precursor
Material and method:
3: Line 92: Include an explanation stating the randomization process, such as "Randomization was performed using a computer-generated sequence with a 1:1:1 allocation ratio, and allocation concealment was ensured using sequentially numbered, opaque, sealed envelopes."
Our answer
We did not use confounding variable because of small sample size/
Mention who conducted the randomization and whether any steps were taken to balance confounding variables.
Include a section describing how the sample size was calculated, such as: "The sample size was estimated using GPower 3.1.2 with a power of 80%, an alpha of 0.05, and an expected effect size of 0.7 based on previous studies on ASD treatments."* If no formal calculation was performed, acknowledge the limitation.
Our answer
All 39 patients, who were clearly diagnosed as autism spectrum disorder aged 7-14 years old, were included in the clinical trial
Miss Mamiko Koshiba who was our (Professor Graduate School of Sciences and Technology for Innovation, Yamaguchi University, Yamaguchi, Japan) conducted the randomization, Because of small sample size, we did not conducted steps taken to balance confounding variables.
Include a section describing how the sample size was calculated, such as: "The sample size was estimated using GPower 3.1.2 with a power of 80%, an alpha of 0.05, and an expected effect size of 0.7 based on previous studies on ASD treatments."* If no formal calculation was performed, acknowledge the limitation.
Our answer
We inserted the sentence in the statistical section marked by yellow color such as The sample size was calculated, such as: "The sample size was estimated using GPower 3.1.2 with a power of 80%, an alpha of 0.05, and an expected effect size of 0.7 based on previous studies on ASD treatments.
4: Line 100 Specify if participants were blinded or if anyone were blinded at each research step (recruitment, intervention, and analysis).
Oue answer
We added the sentences such as participants were blinded or if anyone were blinded at each research step (recruitment, intervention, and analysis) in the Method section marked as yellow colors.
Line 125-128: There are no details about how adverse events (AEs) were monitored and reported.
Our answer
We added the sentence such as “There were no adverse events in the 39 subjects by small doses used in this clinical trial in the last sentence of the result section marked by yellow color
5: "Written informed consent was obtained from the participants', their parents, or both."
Incorrect apostrophe usage in "participants'". "Written informed consent was obtained from participants, their parents, or both."
Our answer
We describe written informed consent as fellow “Moreover, written informed consent was obtained from the participants’, their parents, or both. The study was conducted in accordance with the principles of the Declaration of Helsinki in the method section marked by yellow color.
6: Lines 192:
Blood sampling procedures
How long were blood samples stored? Were they tested in batches or continuously?
Our answer
We stored serum of the blood 10 minutes after blood collection. We tested
30 minutes after blood collection
7: Statistical analysis:
Did the analysis adujsted for baseline ASD severity, age, or gender?
Our answer
We conducted statistical analysis for baseline ASD severity, gender and age.
However, female subjects were very small, We did not assess the effects of gender.
8: Lack of Follow-Up Assessment After 16 Weeks
The study only assesses participants up to 16 weeks, with no long-term follow-up.
Was the improvement maintained after treatment discontinuation?
Our Answer
We assessed the improvement after the end of this clinical trial, About 36 subjects among the 39 subjects were able to reintegrate into society, entering academic high schools.
Discussion:
9: lines 326-330: The discussion highlights the statistical significance of improvements, but not the effect sizes (Cohen's d) or clinical significance.
It is unclear if these improvements are applicable to real-world ASD symptom treatment.
Our answer
We described the sample size in the statistical section as followed marked by yellow color “Sample size was calculated with G*Power 3.12 for the within–between interaction (input parameter: effect size 0.70 with 80% power and 0.05 significance level in a two sided test. The G*Power software supports sample size and power calculation for various statistical methods. This software is helpful for researchers to estimate the sample size and to conduct power analysis.” We inserted the sentences such as “All 36 subjects of the 39 patients were improved in the behavioral and social symptoms in the Results section in our follow up three years examination.” In the efficacy result. Added the sentence such as “Remaining two subjects remained social communication difficulties possibly because of parents` of too much love their children” marked by yellow colors
10: The study claims novel findings regarding RIS-OS effects on motivation and mannerisms but does not compare them with previous ASD risperidone trials.
Our answer
There were no report on the effects of oral risperidone solution. A few studies reported that only three randomized controlled trials of risperidone were identified. Meta‐analysis was possible for three outcomes. Only three randomised controlled trials were identified. Some evidence of the benefits of risperidone in irritability, repetition and social withdrawal were apparent. Risperidone can be beneficial in some features of autism. However there are limited data available from studies with small sample sizes. In addition, there lacks a single standardised outcome measure allowing adequate comparison of studies, and long-term follow up is also lacking (Cochrane Database Syst Re. 2007 Jan 24;2007(1):CD005040.
doi: 10.1002/14651858.CD005040. [Risperidone for autism spectrum disorder.] Authors O S Jesner 1, M Aref-Adib, E Coren
11: The conclusion does not suggest next steps for research, missing an opportunity to guide future ASD studies.
Our answer
Further information on research design is needed to be available in the prominent journal such as World Psychiatry (Impact factor,15,89) linked to this article in the Conclusion section marked by yellow color.
Submission Date
09 February 2025
Date of this review
23 Feb 2025 12:31:41

Reviewer 3 Report
Comments and Suggestions for Authors
1. Abstract: The phrase "These results also validated using in vivo studies" should be revised for clarity. A better alternative would be: "These results were further validated through in vivo studies."
2. Introduction: The sentence "A ton of research has implicated..." is too informal for an academic paper. Consider revising it to "Extensive research has established..." for a more professional tone.
3. The discussion briefly mentions how risperidone interacts with dopamine D2 and serotonin receptors.
4. The manuscript should include a brief discussion comparing the findings with prior studies using risperidone or PUFA supplementation for ASD treatment. Do the current results align with or differ from earlier trials?
5. A simple behavioral analysis could be conducted on a larger cohort of ASD patients to validate the observed effects of risperidone and PUFA supplementation.
6. The study assesses plasma ceruloplasmin and IGF levels, but it would be helpful to analyze their correlation with improvements in ASD symptoms, particularly social responsiveness.
7. Since the study evaluates specific omega-3 to omega-6 ratios, testing a 2:1 or 3:1 ratio (without requiring a long-term study) could provide insights into whether a different formulation yields better results.
8. The study should explicitly discuss potential limitations, such as the lack of a placebo control group and the relatively small sample size. Additionally, suggesting future research directions would enhance the manuscript's impact.
Author Response
Reviewer 3
|
Yes |
Can be improved |
Must be improved |
Not applicable |
|
|
Does the introduction provide sufficient background and include all relevant references? |
( ) |
( ) |
(x) |
( ) |
|
Is the research design appropriate? |
( ) |
( ) |
(x) |
( ) |
|
Are the methods adequately described? |
( ) |
( ) |
(x) |
( ) |
|
Are the results clearly presented? |
( ) |
( ) |
(x) |
( ) |
|
Are the conclusions supported by the results? |
( ) |
( ) |
(x) |
( ) |
Comments and Suggestions for Authors
- Abstract: The phrase "These results also validated using in vivo studies" should be revised for clarity. A better alternative would be: "These results were further validated through in vivo studies."
Ower answer
We added the sentence: “These results were further validated through in vivo studies."
Introduction: The sentence "A ton of research has implicated..." is too informal for an academic paper. Consider revising it to "Extensive research has established..." for a more professional tone.
Our answer
In the introduction section, we described such as “A extensive study on risperidone treatment for children and adolescents with ASD indicated that it significantly improved the restricted, repetitive, and stereotyped patterns of behavior, interests, and activities of children with ASD but did not significantly improve their deficits in social interaction and communication [8]” marked by green color
The discussion briefly mentions how risperidone interacts with dopamine D2 and serotonin receptors.
Our answer
We added the following sentences in the Discission section as follow “The actions of risperidone on brain activity via statistical modeling and pharmacological reversal (risperidone + 5-HT1AR antagonist WAY-100635, risperidone + 5-HT2A/2CR agonist DOI, risperidone + D2R agonist quinpirole). Risperidone, 5-HT1AR agonism with 8-OH-DPAT, 5-HT2AR antagonism with M100907, An important role of 5-HT1AR agonism and 5-HT2AR antagonism in risperidone-induced alterations of delta, beta and gamma oscillations, while D2R antagonism may contribute to risperidone-mediated changes in delta oscillations. The finding provides novel insight into the neural mechanisms for prescribed psychiatric medication targeting the serotonin and dopamine systems in two regions [59]. Therefore, RIS has demonstrated 5-HT1A and 5-HT2A antagonism as well as D2R antagonism [59]. Moreover, risperidone administration resulted in learning deficits during the acquisition phase, indicating a differential role of 5-HT in the acquisition and extinction of an operant conditioning task, suggesting that it may have a dual function in reward encoding, sggesting -HT2A may be encoders of rewards and be involved in the acquisition and extinction of reward encoding [60].
The manuscript should include a brief discussion comparing the findings with prior studies using risperidone or PUFA supplementation for ASD treatment. Do the current results align with or differ from earlier trials?
Our answer
There were no studies on using risperidone or PUFA supplementation for ASD treatment. The present clinical trial revealed the novel findings on the effects of oral risperidone solution (autistic
social motivation) and DHA precursorα-linolenic acid/ARA precursor linolenic acid= 1/1 may be excellent treatment for social motivation in autism spectrum disorder
A simple behavioral analysis could be conducted on a larger cohort of ASD patients to validate the observed effects of risperidone and PUFA supplementation.
Our answer
Yes. A simple behavioral analysis was conducted on a cohort of ASD patients to validate the observed effects of oral risperidone solution and PUFA supplementation.
- The study assesses plasma ceruloplasmin and IGF levels, but it would be helpful to analyze their correlation with improvements in ASD symptoms, particularly social responsiveness.
Our answer
We described the sentences on IGF levels: using risperidone or PUFA supplementation for ASD treatment in the Discussion section as follows: In our study, plasma IGF levels at week 8 were significantly higher in the RIS-OS and awake groups than in the awake groups. RIS has been shown to using risperidone or PUFA supplementation for ASD treatment. increase plasma IGF levels [68], IGF has anti-inflammatory effects on astrocytes and microglia due to the inhibition of blood-brain barrier permeability [68]; further, IGF levels are positively correlated with the ARA precursor linoleic acid [69]. Therefore, the presence of high linoleic acid levels may increase plasma IGF levels [70]. In addition, since RIS has anti-inflammatory activity [71], it promotes the inflammatory effects of plasma IGF levels. This is the first study to reveal that plasma IGF levels are associated with RIS-OS and are positively associated with plasma levels of linoleic acid.
- Since the study evaluates specific omega-3 to omega-6 ratios, testing a 2:1 or 3:1 ratio (without requiring a long-term study) could provide insights into whether a different formulation yields better results.
Our answer
We did not conduct the effects of clinical trial to evaluates specific omega-3 to omega-6 ratios, testing a 2:1 or 3:1 ratio (without requiring a long-term study. However, the findings in the present our study might suggest slight effective to evaluates specific omega-3 to omega-6 ratios, testing a 2:1 or 3:1 omega-3 to omega-6 ratios ratio (without requiring a long-term study)
The study should explicitly discuss potential limitations, such as the lack of a placebo control group and the relatively small sample size. Additionally, suggesting future research directions would enhance the manuscript's impact.
Our answer
We added the sentence such as “
“Limitation
This study included relative small samples the lack of a placebo control group. Need for provide further approaches: Offering specific suggestions for avenues for further investigation. Demonstrate a proactive approach by encouraging future research that addresses the identified more large samples” marked as green color in the end of the Duscussion section
Submission Date
09 February 2025
Date of this review
25 Feb 2025 06:25:28

Round 2
Reviewer 1 Report
Comments and Suggestions for Authors
Most of the previous corrections have been made appropriately.
Please double-check that "- 9.614 -" in the Lower for week 4 in Table 3 is appropriate.
Author Response
【Reviewer1】
Comments and Suggestions for Authors
Most of the previous corrections have been made appropriately.
Please double-check that "- 9.614 -" in the Lower for week 4 in Table 3 is appropriate.
Our Answer
We have corrected an error in the table's description. We have highlighted in light blue. We thank you for your peer review.
Submission Date
09 February 2025
Date of this review
07 Mar 2025 14:53:36
Reviewer 3 Report
Comments and Suggestions for Authors
No more comments
Author Response
【Reviewer 3】
Comments and Suggestions for Authors
No more comments
Our Answer
Thank you for your valuable time in reviewing our paper.
Submission Date
09 February 2025
Date of this review
08 Mar 2025 10:43:53